# Sensitivity Enhancement of a Surface Plasmon Resonance Sensor with Platinum Diselenide

**DOI:** 10.3390/s20010131

**Published:** 2019-12-24

**Authors:** Yue Jia, Zhongfu Li, Haiqi Wang, Muhammad Saeed, Houzhi Cai

**Affiliations:** 1College of Physics and Optoelectronic Engineering, Shenzhen University, Shenzhen 518060, China; onlyjiayue@hotmail.com (Y.J.); lzf_857948523@outlook.com (Z.L.); whq143@outlook.com (H.W.); 2Institute for Advanced Study, Shenzhen University, Shenzhen 518060, China; saeedphysics96@gmail.com

**Keywords:** biosensor, Kretschmann, PtSe_2_, SPR, transition metal dichalcogenides (TMDCs)

## Abstract

The extraordinary optoelectronic properties of platinum diselenide (PtSe_2_), whose structure is similar to graphene and phosphorene, has attracted great attention in new rapidly developed two-dimensional (2D) materials beyond the other 2D material family members. We have investigated the surface plasmon resonance (SPR) sensors through PtSe_2_ with the transfer matrix method. The simulation results show that the anticipated PtSe_2_ biochemical sensors have the ability to detect analytic. It is evident that only the sensitivities of Ag or Au film biochemical sensors were observed at 118°/RIU (refractive index unit) and 130°/RIU, whereas the sensitivities of the PtSe_2_-based biochemical sensors reached as high as 162°/RIU (Ag film) and 165°/RIU (Au film). The diverse biosensor sensitivities with PtSe_2_ suggest that this kind of 2D material can adapt SPR sensor properties.

## 1. Introduction

We have investigated biochemical sensors based on the surface plasmon resonance technique, which has been applied in extraordinary biological and chemical applications, because of their biocompatibility, excellent sensitivity, and accurate detection for pharmaceutics, for instance, in medical diagnostics and enzyme detection [1,2,3,4]. Surface plasmon resonance (SPR) refers to the resonance excitation of the surface plasmon polaritons (SPPs) at the interface of the negative (metal) and positive (dielectrics) constant materials, which is considered as best suited for untagging sensing and real-time monitoring [5,6].

One of the most effective ways for improving the efficiency of sensing devices is to select a material, such as graphene, to optimize sensing functions [7,8,9,10]. The significant properties of transition metal dichalcogenide (TMDC) materials, such as their absorption rate (~5%), which is higher compared to a graphene monolayer (2.3%), entirely different large tunable band gap than the zero band gap of graphene, and large biosense work function in comparison to graphene, are increasingly becoming preferred in biosensing applications. The outstanding electrical, optical, and chemical properties of TMDCs are converting them as promising candidates and accessory materials in comparison to graphene for the future generation of electronic and optic facilities [11,12,13]. TMDC-based SPR sensors are recommended for refractive index sensing and exhibit highly improved sensitivity [14,15].

A series of ultrasensitive photonic crystal fiber based surface plasmon resonance fiber sensors have been made by Md. Saiful Islam et al. proposed polarization-sensitive sensors that showed a relatively high wavelength sensitivity of 25,000 nm/RIU and a high detection limit [16,17,18]. Wu et al. obtained the highest sensitivity of 134.6°/RIU by using a graphene layer for the SPR biosensor to enhance the sensitivity caused by the light absorbed [15,19]. L. Wu and Z. Lin et al. have produced MoS_2_–graphene hybrid structures that are biocompatible and useful in the field of biosensors. Maximum sensitivities of 182°/RIU and 190°/RIU were obtained with 4-layer MoS_2_ and a monolayer of graphene or 6-layer MoS_2_ coatings, respectively, both on surfaces of Al thin film [20,21]. Ouyang et al. reported that PR biosensors using MoS_2_ to improve the sensitivity had the highest sensitivity of 125°/RIU [22].

As single-layer PtSe_2_ exhibits the same pattern of structure as graphene and phosphorene, it also keeps excellent optical and electrical properties that have attracted great attention as a 2D material beyond the predecessor members [10,23,24]. PtSe_2_ is a group ten TMDC with a 1T-phase [25]. The band gap of PtSe_2_ is highly tunable because of its intrinsic quantum confinement effect and strong interlayer interaction. This leads to a type-Ⅱ Dirac semimetal-to-semiconductor transition when going from bulk to few-layer form and exhibits the largest band gap of ~1.2 eV for monolayer (ML) PtSe_2_ (from theoretical prediction) [26,27]. Moreover, different types of stress can be applied on PtSe_2_, which can be modulated easily [28]. Furthermore, less toxicity and chemical stability has been investigated in PtSe_2_ sensing applications [23,29,30]. TMDC PtSe_2_ monolayers not only possess a significant thermoelectric character, having semiconductor properties, but it also has an outstanding optoelectronic property. To the best of our knowledge, few systematic studies have been performed on the optic and electric properties of PtSe_2_.

In the visible light region, Ag and Au are considered ideal candidates as metallic films [4]. Ag films usually give a sharper peak compared Au and offer better sensitivity, which is used for enhancing biosensor sensitivity [31]. In the reaction system, however, the Au film shows stability, strong adhesion with the glass, and it produces no reaction with inorganic ions. As Au is not susceptible to oxidation, and it usually does not react with most chemicals, it is usually used as metallic film in sensors [32]. Here, Ag and Au film is used as a metallic layer coating on top of the BK7 substrate in SPR biological systems. The 2D material PtSe_2_ might be used as the protective layer adjoining the biomolecular recognition elements to prevent oxidation and increase biomolecule adsorption. In this paper, the 2D material PtSe_2_ is used in SPR biochemical sensors to enhance the sensitivity and retain chemical stability. Here, Kretschmann’s attenuated total reflection (ATR) configuration has been selected as the SPR sensor structure [33]. Kretschmann’s configuration is one of the most basic SPR configurations of biosensors and is usually composed of a coupling prism and metallic film. Biomolecules and metallic membranes can interact with each other, through which biomolecules can be detected. We have used Ag or Au to cover, on the basis of the optical coupling prism, BK7 glass to be used as the coupling prism, and PtSe_2_ covers the metallic layer in this structure [34,35].

## 2. Calculation Models and Methods

Z. Lou and other authors provided their synthesized nanoparticle−organic clusters (NOCs) as signal amplification reagents permitting an SPR signal four times higher than that of the sandwich format [36,37,38]. Here, our design presents a proposed structure having a high-sensitivity sensor separately containing PtSe_2_, as shown in Figure 1. In these SPR structures, the Ag or Au film is installed on top of the BK7 coupling prism, and the PtSe_2_ layer is the biomolecular recognition element coated on the Ag/Au film surface. The size of the thickness of the PtSe_2_ monolayer is 0.375 nm [27]. We have used BK7 glass as a coupling prism. As the first-rank metal for SPP, the Au film or Ag film is chosen at 50 nm. The BK7 glass refractive index can be calculated using the following equation [39]:(1)nBK7=(1.03961212λ2λ2−0.00600069867+0.231792344λ2λ2−0.0200179144+1.03961212λ2λ2−103.560653+1)12

The refractive index of Ag or Au can be expressed from the Drude–Lorentz model [40]:(2)nm=εm=[1−λ2λcλp2(λc+iλ)]12
where *λ_c_* and *λ_p_* represent the collision and plasma wavelengths. The values of *λ_c_* and *λ_p_* for Ag are 1.7614 × 10^−5^ m and 1.4541 × 10^−7^ m, whereas for Au they are 8.9342 × 10^−6^ m and 1.6826 × 10^−7^ m, respectively. In order to further enhance the sensibility of the designed biochemical sensors, we have covered the metal Ag or Au film surface with PtSe_2_ to impede the metal being oxidized.

Experimental modulating is used to get the refractive indices as a variable quantity of PtSe_2_, which is composed of real and imaginary parts [41]. The real and imaginary parts of the complex dielectric function of PtSe_2_ for a wavelength at 633 nm are exhibited in Figure 2. The expression *n_s_* = 1.33 + Δ*n* is presented as the refractive index in the sensitive medium, and Δ*n* represents refractive index change in sensitive medium as a result of biological action or a chemical reaction.

The transfer matrix method has been used [42] to investigate the reflectivity of the incident TM-polarized light of the *N*-layer model, as the matrix method is accurate and without approximations. MATLAB was used to compute the analogous SPR modulation. In these PtSe_2_ biosensors, all layers are piled in the direction of the vertical BK7 glass coupling prism, and each layer is named by the refractive index (*n_k_*), thickness (*d_k_*), and dielectric constant (*ε_k_*), respectively. The resonance angle is the minimum reflectance corresponding to the incident angle. The tangential field of the first boundary *Z* = *Z*_1_ = 0 is correlated with the tangential field of the final boundary *Z* = *Z*_*N*−1_:(3)$$\left[\begin{array}{c} U_{1}\\ V_{1} \end{array}\right]=M\left[\begin{array}{c} U_{N-1}\\ V_{N-1} \end{array}\right]$$
where *V* and *U* represent the components of magnetic and electric fields at the limiting surface. *M* and *k* present the characteristic matrix of the composite architecture and the *k*th layer in the *N*-layered model, respectively. P-polarized light is made up of:(4)$$\prod_{K=2}^{N-1}M_{K}=\left[\begin{array}{cc} M_{11} & M_{12}\\ M_{21} & M_{22} \end{array}\right],$$
(5)$$M_{K}=\left[\begin{array}{cc} \cos\beta_{K} & \frac{-\sin\beta_{K}}{q_{k}}\\ -{\mathrm{iq}}_{k}\sin\beta_{k} & \cos\beta_{k} \end{array}\right],$$
(6)qk=(μkδk)12cosθk=(εk−n12sinθ12)12εk,
(7)βk=2πλnkcosθk(Zk−Zk−1)=2πdkλ(εk−n12sinθ12)12.

After mathematical simplifications, we get the p-polarized light *N*-layer complex reflection coefficient *r_p_*, and the corresponding amplitude reflection coefficient (*R_p_*) might be obtained by the square of *r_p_*:(8)$$r_{p}=\frac{\left(M_{11}+M_{12}q_{5}\right)q_{1}-\left(M_{21}+M_{22}q_{5}\right)}{\left(M_{11}+M_{12}q_{5}\right)q_{1}+\left(M_{21}+M_{22}q_{5}\right)},$$
(9)$$R_{p}={\left|r_{p}\right|}^{2}.$$

The transformation of sensing medium in refractive index (Δ*n*) can initiate the alteration of the resonance angle (Δ*θ*), and the sensitivity can be indicated as SR1 = Δ*θ*/Δ*n* [43].

The detection accuracy (DA) is the ratio in the reflectance curve of shift in resonance angle (*∆θ_res_*) to the full width at half-maximum (FWHM):(10)DA=Δθ/FWHM

The figure of merit (FOM) is the ratio in the reflectance curve of sensitivity (S) to the FWHM [44]:(11)FOM=S/FWHM

## 3. Results and Discussion

Because of the low refractive index of BK7 glass, it was chosen as the coupling prism in these expected biochemical sensors. For the conventional SPR, the Kretschmann geometry biochemical sensor mostly offers a simplex metallic layer to survey SPP, as illustrated in Figure 3. The values of the sensitivity of the conventional Ag or Au metal were 118°/RIU and 130°/RIU, as shown in Figure 3a,c, respectively, by using BK7 glass prism. However, this value is not satisfactory for biochemical sensor sensitivity [45,46]. In this paper, we have developed SPR biochemical sensors by using 2D TMDC PtSe_2_ materials to enhance the sensitivities of the sensors.

The addition of the 2D TMDC PtSe_2_ material between the metallic (Au or Ag) film and sensing medium in the structure of biosensors increased the sensitivities. The results clearly suggest a large improvement in the sensitivities when the PtSe_2_ monolayer covers the previous structure, as shown in Figure 3b,d. For the Ag film or Au film, we have obtained sensitivities equal to 162°/RIU and 165°/RIU at the layers accumulating 16 and 12 layers, respectively.

In future experiments, 2D PtSe_2_ nanosheets can be prepared by the liquid-phase exfoliation (LPE) method, and then the nanosheets will be coated on the Ag or Au thin film (~50 nm) with the BK7 substrate by using the spin coating method [47]. The SPR sensors can be constructed by connecting the PtSe_2_–Ag/Au–BK7 multilayer films and the BK7 prism with the refractive index matching solution.

Based on these biochemical sensor results, the 2D TMDC PtSe_2_ material might increase sensitivities. The reflectivity of PtSe_2_ layers changed with the variation in the incident angle, which is shown in Figure 4a,c, illustrating that sensitivities varied with different layers of PtSe_2_-based biochemical sensors from *n_s_* = 1.33 to *n_s_* = 1.37 to the sensing medium refractive index. The reflectivity changes with the incident angle are shown in Figure 4b,d. It is clear from Figure 4b,d that, with the increase of refractive indices of the sensitive media, the angles of resonance of PtSe_2_ sensors at 2.0 nm thickness shifted to a higher angle of incidence. The resonance angles were 67.54°, 68.13°, 68.75°, 69.39°, and 70.06° when *n_s_* = 1.33, *n_s_* = 1.34, *n_s_* = 1.35, *n_s_* = 1.36, and *n_s_* = 1.37 for Ag film, respectively. Whereas, the resonance angles for Au film were 70.33°, 71.01°, 71.73°, 72.49°, and 73.28° when *n_s_* = 1.33, *n_s_* = 1.34, *n_s_* = 1.35, *n_s_* = 1.36, and *n_s_* = 1.37, respectively.

Figure 5a presents distributions of the electric field for the proposed SPR sensors for PtSe_2_ material on Ag or Au film, respectively. It has been investigated that coating PtSe_2_ on the Ag film might enhance the electric field at the sensor/sensing medium interface, where PtSe_2_ gets the maximum electric field strength, since PtSe_2_ appears in the middle and is located at the last single layer of PtSe_2_.

The changes of sensitivities for PtSe_2_ on Ag and Au surface biosensors are plotted in Figure 5b. It can be seen that the values of the highest sensitivities of PtSe_2_ occur at 162°/RIU and 165°/RIU, respectively. Then, the sensitivity curves decrease with an increasing PtSe_2_ layer thickness, causing the gold or silver film coating on the PtSe_2_ nanosheet overlayer to affect the evanescent electric field distribution, which, in turn, has an impact on the decay length [47]. Figure 5c shows the electric field distributions of 2.0 nm PtSe_2_ on Ag film and Au film biochemical sensors, separately. The refractive indices of the inductive media changed within the range of 1.33 to 1.37, and the change in the electric field of graphene/sensing medium interface has been clearly observed. It was found that the designed PtSe_2_-based biochemical sensors are sensitive when detecting small changes in sensitive media. A major change of the surface wave characteristics might occur as a result of small changes in the refractive index of sensitive medium close to the interface, which may produce change in the electric field. Figure 5d gives the change of FOM for this PtSe_2_ biosensor. When the thickness of the 2D PtSe_2_ material was from 0 to 7 nm, and the change of width was kept from 1° to 3°, we calculated the FOM using Equation (11). From this calculation, we can determine that the FOM decreases with the increase of thickness on both Ag and Au films.

Table 1 presents the information about change in thickness of PtSe_2_, layer numbers (*L*), highest sensitivity (*S*), minimum resonance angle (*θ_min_*) at *n_s_* = 1.33, and resonance angle change (Δ*θ*) from *n_s_* = 1.330 to *n_s_* = 1.335. The smallest reflectivity (*R_min_*), FWHM, and FOM were at *n_s_* = 1.33, and DA was at *n_s_* = 1.33 to *n_s_* = 1.34 in these PtSe_2_ biochemical sensors on Ag or Au films. The thicknesses of Ag and Au were set to 50 nm. We set the range of the incident angle from 0° to 90°. The thicknesses of PtSe_2_ layers came from reference [41], and layer numbers were calculated from the thickness of the PtSe_2_ monolayer, which was 0.375 nm. The reflectivity curve steps forward at a higher angle with an increase in the number of PtSe_2_ layers. Once the PtSe_2_ layers are augmented to the optimized value, maximum sensitivity can be obtained. For PtSe_2_ on Ag and Au, the sensitivities achieved maximum values as the layers accumulated to 16 and 12, respectively. As a result of certain restrictions on the angle range, there was a direct relation of the variation of the resonance angle with the sensitivity (i.e., as the variation of the resonance angle lowered, the sensitivity decreased after exceeding the optimized PtSe_2_ layers). In the end, we are also aware that most high-sensitivity sensors are accompanied by an increase in SPR linewidth and a reduction in figure of merit (FOM), and a higher FOM usually means a lower sensitivity [48]. Table 1 also lists the FWHM, DA, and FOM data for these PtSe_2_ biosensors.

When the dielectric refractive index change is Δ*n*, the reflection angle of the prism with a lower refractive index gave a larger result, which means that when the refractive index of the prism is low, a relatively higher sensitivity can be obtained [20]. Therefore, higher sensitivity can be obtained by using a prism with a low refractive index. The refractive indexes of silicon and BK7 were calculated as 3.420 and 1.515 at λ = 633 nm, respectively [15,49]. Therefore, to improve the sensitivity, BK7 was chosen as the coupling prism in the proposed architecture.

Table 2 gives the information about the optimized layers of different architectures, changes in resonance angle (Δθ), highest sensitivity (S), and FOM from another two previous papers in our laboratory. Comparing with these data of graphene, MoS_2_, black phosphorous heterojunction, and diversification architectures [51], PtSe_2_-based biosensors did not have the highest sensitivity. However, this high sensitivity and simple biosensor structure makes it easy to achieve commercial production.

## 4. Conclusions

On the basis of the Kretschmann configuration, we have theoretically investigated a new type of biosensor structure with the insertion of a PtSe_2_ layer. In this paper, SPR biochemical sensors utilizing 2D TMDC PtSe_2_ have been designed and emulated to improve the sensitivities. Moreover, the various characters of PtSe_2_ on different metal substrates have been discussed in detail. In order to obtain a high sensitivity, we used BK7 as the coupling prism in this structure because of its low refractive index. Ag or Au films were covered with PtSe_2_ to enhance the sensitivities as well as the protective layers. It was found that they enhanced the sensitivities in the advised biochemical sensors, and the sensitivities of PtSe_2_ were 162°/RIU and 165°/RIU for Ag and Au substrates at PtSe_2_ layers 16 and 12, respectively. We are convinced that these kinds of sensors can have potential applications in environmental monitoring, biological detection, medical diagnosis, chemical examination, and so on.

## Figures and Tables

**Figure 1 sensors-20-00131-f001:**
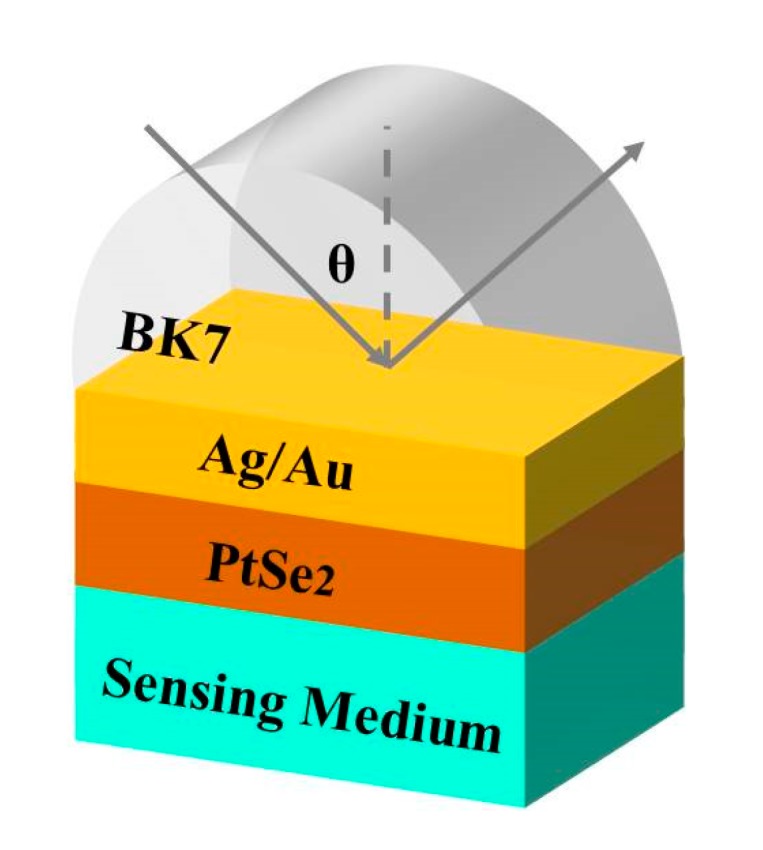
Sketch diagram of the PtSe_2_ surface plasmon resonance (SPR) biochemical sensor to enhance the sensitivity.

**Figure 2 sensors-20-00131-f002:**
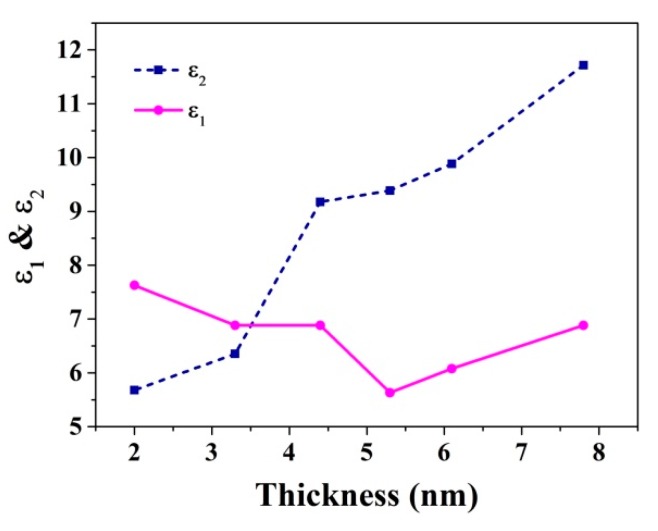
Real and imaginary parts of the complex dielectric function for PtSe_2_ at a wavelength of 633 nm.

**Figure 3 sensors-20-00131-f003:**
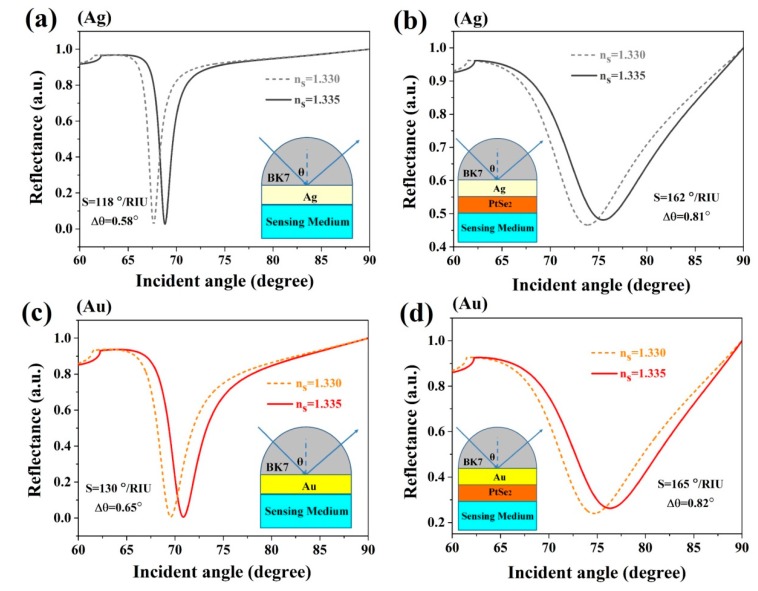
Variation of reflectivity with incident angles for (**a**),(**c**) the conventional biochemical sensor based on simplex Ag or Au film; and (**b**),(**d**) the proposed biochemical sensors with PtSe_2_ on Ag or Au, respectively.

**Figure 4 sensors-20-00131-f004:**
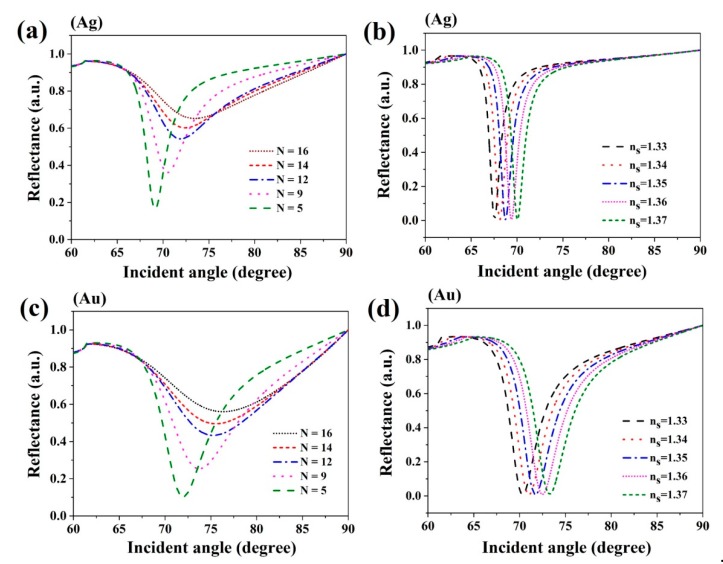
(**a**,**c**) The reflectances of the PtSe_2_ biosensor on Ag film and Au film change with different numbers of PtSe_2_ layers, respectively; (**b**,**d**) the reflectance of PtSe_2_ biosensor with 2.0 nm PtSe_2_ varies with the refractive indices of the sensing medium on Ag film and Au film, respectively.

**Figure 5 sensors-20-00131-f005:**
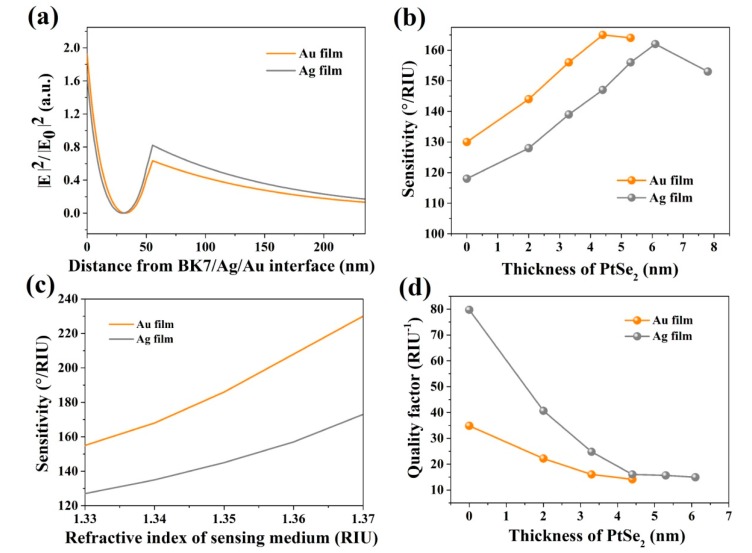
(**a**) Electric field distributions for the PtSe_2_ on Ag film and Au film sensors; (**b**) Variation of sensitivity with respect to PtSe_2_ in different thicknesses; (**c**) The sensitivities of the proposed biochemical sensors with 2.0 nm PtSe_2_ vary with the refractive indices of sensing medium on Ag film and Au film, respectively; (**d**) Variation of quality factors with respect to the thickness of PtSe_2_ varying from 0 to 7 nm of PtSe_2_ biosensor.

**Table 1 sensors-20-00131-t001:** Change in thickness of PtSe_2_, layer numbers (*L*), highest sensitivity (*S*), minimum resonance angle (*θ_min_*) at *n_s_* = 1.33, and resonance angle change (*Δθ*) from *n_s_* = 1.330 to *n_s_* = 1.335. The smallest reflectivity (*R_min_*), full width at half-maximum (FWHM), and figure of merit (FOM) is at *n_s_* = 1.33, and detection accuracy (DA) is at *n_s_* = 1.33 to *n_s_* = 1.34 for the PtSe_2_ SPR biochemical sensors.

	PtSe_2_	Thickness (nm)	*L*	*S* (◦/RIU)	*θ**_min_* (Degree)	Δ*θ* (Degree)	*R_min_* (a. u.)	FWHM	DA	FOM
Ag	Without PtSe_2_	0	0	118	67.64	0.59	0.0269	1.48	0.797	79.72
	PtSe_2_	2.0	5	128	69.14	0.64	0.1096	3.15	0.4063	40.63
		3.3	9	139	70.49	0.69	0.2951	5.61	0.2477	24.77
		4.4	12	147	71.90	0.73	0.4837	9.17	0.1603	16.03
		5.3	14	156	72.84	0.78	0.5487	9.96	0.1566	15.66
		6.1	16	162	73.86	0.81	0.6043	10.85	0.1493	14.93
Au	Without PtSe_2_	0	0	130	69.57	0.65	0.0063	3.73	0.3485	34.85
	PtSe_2_	2.0	5	144	71.40	0.72	0.0555	6.49	0.2218	22.18
		3.3	9	156	73.01	0.78	0.1808	9.74	0.1601	16.01
		4.4	12	165	74.69	0.82	0.3515	11.68	0.1412	14.12

**Table 2 sensors-20-00131-t002:** The optimized layers, change in resonance angle (Δθ), highest sensitivity (S), and reference.

Layers in the Structure	Δ*θ* (Degree)	*S* (◦/RIU)	Reference
Ag, 5 nm black phosphorus	0.90	181	[50]
Ag, 5 nm BP, 5 L graphene	1.08	217	[50]
Ag, 5 nm BP, 1 L MoS_2_	1.09	218	[50]
Ag, 5 nm BP, 1 L WS_2_	1.18	237	[50]
Au, 4 L MoS_2_, Au, 1 L graphene	0.91	182	[20]
Ag, 6.1 nm PtSe_2_	0.81	162	This work
Au, 4.4 nm PtSe_2_	0.82	165	This work

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
