# Peer review of "Sensitivity Enhancement of a Surface Plasmon Resonance Sensor with Platinum Diselenide"

_sensors, 2019, doi:10.3390/s20010131_

Round 1

Reviewer 1 Report

In this manuscript, the authors investigate the SPR sensors through PtSe2 with transfer matrix method. The simulation results show that the anticipated PtSe2 contained biochemical sensors have the ability to detect analytic with improved sensitivity. The obtained results are very important in this field. And I suggest to accept this paper after minor revisions as below:

In the first sentences in paragraph 2 in the introduction section, the authors use “optimize functions”, “work functions” to discuss the advantages of using TMDCs. However, they failed to descript what these functions exactly are.

The authors should discuss the difference between the methods they design in this work and the traditional “sandwich” detection methods (such as Int. J. Mol. Sci., 20(3), 741; Nanomaterials, 8(2), 107; Analytical Chemistry, 89(24):13472-13479), since both of them are based on “layer by layer” mechanism and are proved to have the abilities to improve the sensitivity of SPR. By doing this, the authors can prove the importance of their work.

The equation (1) is in error display.

In Line 145-147 on Page 5, the authors state that “In future ...”. Do they have any practical example? Some references are necessary.

Do different ways for assembling of PtSe2 layer by layer, affect the simulation results?

The authors should explain the reason that further increasing the thickness will induce the decrease of sensitivity as shown in Figure 5B.

The authors should explain how they determine the thickness values during their calculation as shown in Table 1.

Since the authors always compare the sensitivity improvement ability between PtSe2 and graphene, some reference is necessary, such as Colloids and Surfaces B, 157: 31-39.

More carful check of the grammar is needed.

Author Response

Reviewer 1

In the first sentences in paragraph 2 in the introduction section, the authors use “optimize functions”, “work functions” to discuss the advantages of using TMDCs. However, they failed to descript what these functions exactly are.

Answer: I added more accurate words referring to specifics. The sentences become “One of the most effective ways for improving efficiency of sensing device is to select a material as graphene to optimize sensing functions [7-10]. The significant properties of transition metal dichalcogenides (TMDCs) materials such as absorption rate (~5%), which is higher compared to graphene monolayer (2.3%), entirely different large tunable band gap than zero band gap graphene, and large biosense work function in comparison to graphene, are increasingly becoming preferred in biosensing applications.”

The authors should discuss the difference between the methods they design in this work and the traditional “sandwich” detection methods (such as Int. J. Mol. Sci., 20(3), 741; Nanomaterials, 8(2), 107; Analytical Chemistry, 89(24):13472-13479), since both of them are based on “layer by layer” mechanism and are proved to have the abilities to improve the sensitivity of SPR. By doing this, the authors can prove the importance of their work.

Answer: I cited these three references and took a compare, shown as “Z. Lou etc. provided their synthesized nanoparticle−organic clusters (NOCs) as signal amplification reagents permitting a 4 times higher SPR signal comparing with sandwich format [36-38]. Here, our design is presenting a proposed structure having high sensitivity sensor separately containing PtSe2 as shown in Fig. 1.”

Yuan, C.; Lou, Z.; Wang, W.; Yang, L. and Li, Y.; Synthesis of Fe3C@C from Pyrolysis of Fe3O4-Lignin Clusters and Its Application for Quick and Sensitive Detection of PrPSc through a Sandwich SPR Detection Assay. Int. J. Mol. Sci. 2019, 20, 741. Lou, Z.; Han, H.; Mao, D.; Jiang, Y.; and Song, J.; Qualitative and Quantitative Detection of PrPSc Based on the Controlled Release Property of Magnetic Microspheres Using Surface Plasmon Resonance (SPR). Nanomaterials 2018, 8, 107. Lou, Z.; Han, H.; Zhou, M.; Wan, J.; Sun, Q.; Zhou, X. and Gu, N.; Fabrication of Magnetic Conjugation Clusters via Intermolecular Assembling for Ultrasensitive Surface Plasmon Resonance (SPR) Detection in a Wide Range of Concentrations. Anal. Chem. 2017, 89, 13472−13479.

The equation (1) is in error display.

Answer: I modified the equation.

In Line 145-147 on Page 5, the authors state that “In future ...”. Do they have any practical example? Some references are necessary.

Answer: I cited more references.

Xiong, X.; Plasmonic interface modified with graphene oxide sheets overlayer for sensitivity enhancement. ACS Appl. Mater. Interfaces. 2018, 10, 34916-34923.

Do different ways for assembling of PtSe2 layer by layer, affect the simulation results?

Answer: This is for application aspect. For simulation aspect is no different, because the required parameters are the same no matter what ways to manufacture. But in the real application, different manufacture will produce the different productions, even you want the same final design, the parameters will change a little bit depends on the last productions, and the result maybe not the same.

The authors should explain the reason that further increasing the thickness will induce the decrease of sensitivity as shown in Figure 5B.

Answer: I rewrite as “Then the sensitivity curves decrease with the increasing of PtSe2 layer thickness, causing the gold or silver film coating on PtSe2 nanosheet overlayer will affect the evanescent electric field distribution, which in turn has an impact on the decay length [47].” In the paper.

Xiong, X.; Plasmonic interface modified with graphene oxide sheets overlayer for sensitivity enhancement. ACS Appl. Mater. Interfaces. 2018, 10, 34916-34923.

The authors should explain how they determine the thickness values during their calculation as shown in Table 1.

Answer: These thicknesses are come from the reference [41. Xie J. et al. Optical properties of chemical vapor deposition-grown PtSe2 characterized by spectroscopic ellipsometry, 2D Mater. 2019, 6, 035011] for more accurate calculation. And I also add the corresponding PtSe2 layer number.

We changed the text as “The thicknesses of PtSe2 layers are come from reference [41], and layer numbers were calculated from the thickness of PtSe2 monolayer which is 0.375nm.”

Since the authors always compare the sensitivity improvement ability between PtSe2 and graphene, some reference is necessary, such as Colloids and Surfaces B, 157: 31-39.

Answer: I added this reference.

Lou, Z.; Wan, J.; Zhang, X.; Zhang, H.; Zhou, X.; Cheng, S.; Gu, N.; Quick and sensitive SPR detection of prion disease-associated isoform(PrPSc) based on its self-assembling behavior on bare gold film andspecific interactions with aptamer-graphene oxide (AGO). Colloids and Surfaces B: Biointerfaces, 2017, 157, 31–39.

More carful check of the grammar is needed.

Answer: I modified the English grammar.

Reviewer 2 Report

Dear Authors,

I think that your draft needs some major improvement before considering the publication in this Journal.

In general, please consider these major issues:

1) I suggest that you describe with greater depth the pros of using PtSe2 instead of Au or Ag alone. In my opinion, the increase of about 30 deg/RIU can not be so relevant if this implies that the plasmonic dip is less sharp, because the shift can be difficult to detect: when you implement your sensor in real world, you have to take into account the sensitivity of your measurement set up.

2) Your draft needs a deep and accurate revision of the English form, at the level of single nouns (e.g., fraternity (35), mental (66), ingredients (111), persuade (135), etc.).

Furthermore, please consider also the following points:

3) some paragraphs are oddly spaced, e.g., r. 49-60.

4) r. 83: how PtSe2 can be defined as biomolecular recognition element? I suppose that you are designing an optical affinity biosensor, so what kind of probe/target binding are you planning to use? Are you aiming to functionalize PtSe2 with probes? And if so, which kind of probes? Please clarify your target application.

5) Fig. 4 b-d: please describe which kind of detection you can implement by sensing medium refractive index variations between 1.33 and 1.37, also with some literature example if necessary.

6) Fig. 5b: from these data it seems that your sensitivity peaks correspond to PtSe2 layers with thicknesses of 4 and 6 nm. Please clarify why did you adopt the 2 nm thickness in Fig. 4.

7) rr. 218-219: from my perspective, these aspects are extremely important. Please describe in greater detail your claim related to the easiness of PtSe2-based sensors commercial production with respect to Ag- and Au-based ones.

Author Response

Reviewer 2

1) I suggest that you describe with greater depth the pros of using PtSe2 instead of Au or Ag alone. In my opinion, the increase of about 30 deg/RIU can not be so relevant if this implies that the plasmonic dip is less sharp, because the shift can be difficult to detect: when you implement your sensor in real world, you have to take into account the sensitivity of your measurement set up.

Answer: I rewrite as “Then the sensitivity curves decrease with the increasing of PtSe2 layer thickness, causing the gold or silver film coating on PtSe2 nanosheet overlayer will affect the evanescent electric field distribution, which in turn has an impact on the decay length [47].” In the paper.

Xiong, X.; Plasmonic interface modified with graphene oxide sheets overlayer for sensitivity enhancement. ACS Appl. Mater. Interfaces. 2018, 10, 34916-34923.

2) Your draft needs a deep and accurate revision of the English form, at the level of single nouns (e.g., fraternity (35), mental (66), ingredients (111), persuade (135), etc.).

Answer: I modified the English grammar.

Furthermore, please consider also the following points:

3) some paragraphs are oddly spaced, e.g., r. 49-60.

Answer: Thank you for the carefully check, I adjusted them.

4) r. 83: how PtSe2 can be defined as biomolecular recognition element? I suppose that you are designing an optical affinity biosensor, so what kind of probe/target binding are you planning to use? Are you aiming to functionalize PtSe2 with probes? And if so, which kind of probes? Please clarify your target application.

Answer: We will make the PtSe2 biosensor as the following progress: First, the glass slides were immersed in an ultrasonic bath for several minutes for cleaning. In the subsequent metal deposition process, a gold or silver film (50 nm) will be successively deposited onto the slides by a vacuum evaporating method. Then to coat the prepared PtSe2 alcohol suspension onto the metallic film and form a PtSe2 modified SPR chip. The PtSe2 alcohol suspension was dropped directly on the surface of the gold film and allowed to stand at room temperature for 10 h and the ethanol was naturally evaporated. The results show that a certain thickness of the PtSe2 film can be firmly attached to the gold layer for further application.

5) Fig. 4 b-d: please describe which kind of detection you can implement by sensing medium refractive index variations between 1.33 and 1.37, also with some literature example if necessary.

Answer: Water refractive index is 1.33 and the target detected substance refractive index is between 1.33 and 1.37. So we calculate the refractive index from 1.33 to 1.37.

6) Fig. 5b: from these data it seems that your sensitivity peaks correspond to PtSe2 layers with thicknesses of 4 and 6 nm. Please clarify why did you adopt the 2 nm thickness in Fig. 4.

Answer: Thanks for the acuminous observational ability. That is true the sensitivity peasks for PtSe2 layer thickness are around 4 nm and 6 nm. But we use 2 nm thickness for exhibition in Fig. 4. cause we always using the starting layer for demonstrate. And in this case for accurate calculate we use the original start layer 2 nm for compute.

7) rr. 218-219: from my perspective, these aspects are extremely important. Please describe in greater detail your claim related to the easiness of PtSe2-based sensors commercial production with respect to Ag- and Au-based ones.

Answer: This PtSe2 biosensor Kretschmann configuration is very simple, and too easy to achieve the preparation process. These characters look forward to celerity commercial implement.

Round 2

Reviewer 2 Report

Dear Author,

thank you for your redrafting efforts. I think that a clarification is needed before I can recommend the publication of your new draft in this valuable Journal.

Referring to my past review, I will try to rephrase that issues:

5) in your answer you state that "target detected substance refractive index is between 1.33 and 1.37". Can you please make few example of such "target detected substance"? What kind of detectable substances can vary the medium refractive index between 1.33 and 1.37, e.g., metal ions, pesticides, drugs? Please define what kind of target substances you can detect in water with your sensor. 

Best regards.

Author Response

Thank you so much for review my article. I am also very interesting on the next step experiment for application. For our experiment in the lab, we all use environment-friendly materials. So now most target detected substance are dissolved in water. For example, heavy metal ions we solve in distilled water, RNA we solve in DEPC water, and the refractive index for water solvent is around 1.33.